# Dioscin Alleviates Cisplatin-Induced Mucositis in Rats by Modulating Gut Microbiota, Enhancing Intestinal Barrier Function and Attenuating TLR4/NF-κB Signaling Cascade

**DOI:** 10.3390/ijms23084431

**Published:** 2022-04-17

**Authors:** Shengzi Jin, Tongxu Guan, Shuang Wang, Mengxin Hu, Xingyao Liu, Siqi Huang, Yun Liu

**Affiliations:** 1College of Veterinary Medicine, Northeast Agricultural University, Harbin 150030, China; jinsz1997@163.com (S.J.); guantongxu@yeah.net (T.G.); wangshuang@neau.edu.cn (S.W.); 13149609373@163.com (M.H.); xingyaoliu@126.com (X.L.); HuangSiQihhh@163.com (S.H.); 2Heilongjiang Key Laboratory for Laboratory Animals and Comparative Medicine, Northeast Agricultural University, Harbin 150030, China

**Keywords:** dioscin, cisplatin, mucositis, gut microbiota, intestinal barrier, TLR4-MyD88-NF-κB

## Abstract

Cisplatin-based chemotherapy causes intestinal mucositis, which causes patients immense suffering and hinders the process of cancer treatment. Dioscin is a natural steroid saponin that exhibits strong anti-inflammatory and immunomodulatory properties. Herein, we investigate the protective effect of dioscin on cisplatin induced mucositis in rats from the perspective of gut microbiota and intestinal barrier. We established a rat model of intestinal mucositis by tail vein injection of cisplatin, and concurrently treated with dioscin oral administration. Parameters, such as body weight, diarrheal incidence, and D-Lactate levels, were assessed in order to evaluate the effects of dioscin on intestinal mucositis in rats. Furthermore, biological samples were collected for microscopic gut microbiota, intestinal integrity, and immune inflammation analyses to elucidate the protective mechanisms of dioscin on intestinal mucositis. The results revealed that administration of dioscin significantly attenuated clinical manifestations, histological injury and inflammation in mucositis rats. Besides this, dioscin markedly inhibited the gut microbiota dysbiosis induced by cisplatin. Meanwhile, dioscin partially alleviated junctions between ileum epithelial cells and increased mucus secretion. Moreover, dioscin effectively inhibited the TLR4-MyD88-NF-κB signal transduction pathway and reduced the secretion of subsequent inflammatory mediators. These results suggested that dioscin effectively attenuated cisplatin-induced mucositis in part by modulating the gut microflora profile, maintaining ileum integrity and inhibiting the inflammatory response through the TLR4-MyD88-NF-κB pathway.

## 1. Introduction

Cisplatin (cis-diamminedichloroplatinum II, CDDP), the best and first metal-based chemotherapeutic drug [1], is one of the most commonly used anticancer agents without a doubt, and is used for the treatment of solid cancers, such as prostate cancer, ovarian cancer, head and neck cancer, bladder and lung cancer [2]. However, the non-specific effects of CDDP, which interfere with the replication of DNA by crosslinking DNA, can have multiple adverse effects [3]. In particular, CDDP shows high activity in the rapidly proliferating intestinal villi cells, thereby causing intestinal mucosal damage [4]. Approximately 40% of chemotherapy patients suffer from gastrointestinal mucositis [5], which has further detrimental effects, such as decreased quality of life, longer hospital stays, higher medical costs, reduced dosages of chemotherapy, or even cessation of treatment [6]. Recent research has demonstrated a significant compositional and functional imbalance in the gut microorganisms after CDDP treatment, contributing to further impairments of the gut structure and function [7,8]. 

Our understanding of the distribution and function of gut microbes has significantly improved over the past decade. In the regulation of intestinal homeostasis, commensal bacteria play an important role and have a certain protective effect on the integrity of the intestine [9]. Their interactions with Toll-like receptors (TLRs), as well as subsequent activation of the NF-κB signaling pathway, regulate immune responses. In addition, commensal bacteria regulate intestinal barrier function, notably by modulating the expression and distribution of tight junction proteins [10,11]. Additionally, the mucus layer is another protective layer that contributes to intestinal integrity, and which is influenced by gut bacteria [12]. Based on basic and clinical studies, the intestinal microbiota may play an important role in the pathogenesis of chemo-induced mucositis [8]. Consequently, restoring gut microbiota dysbiosis may prove to be an effective method of preventing and treating cisplatin-induced mucositis.

Traditional Chinese medicines have been used to prevent human diseases for thousands of years. Studies have shown that many common saponins in diet and traditional Chinese medicine alleviate CDDP-induced intestinal mucositis [4,13]. Dioscin (Dio) is a naturally occurring steroidal saponin, which is isolated from various vegetables and herbs [14]. Dioscin (Dio) is capable of effectively treating cancers [15,16,17,18] and has protective effects on a variety of organs, including the heart, liver, and kidneys, and plays an important role in the prevention and treatment of many diseases [19,20,21]. Previous studies have demonstrated that Dio exerts protective effects against intestinal ischemia/reperfusion injury as well as against cisplatin-induced inflammatory kidney injury [21,22]. In addition, Dio exhibits sustained-release pharmacokinetic characteristics after oral administration. The reason for this is that it is capable of being absorbed by the intestinal wall, and this intestinal retention ability may allow it to continue to be utilized by gut microbes [23]. Nevertheless, no studies are available on the effects of Dio on gastrointestinal mucosal immunity or regulation of intestinal microbes. Therefore, the purpose of this study was to investigate whether dioscin protected against cisplatin-induced mucositis by modulating the gut microbiota, or by restoring intestinal barrier function and reducing ileum inflammation.

## 2. Results

### 2.1. Dio Alleviated CDDP-Induced Mucositis in Rats

Loss of body weight and diarrhoea are common phenomena after CDDP treatment. They are also basic indicators of CDDP entertoxicity [24]. The single injection of CDDP (6 mg/kg) rats exhibited body weight loss, accompanied by diarrhea, compared with control or Dio only-treated rats (*p* < 0.01). However, Dio (60 mg/kg) significantly attenuated disease symptoms in CDDP treated rats, including weight loss (*p* < 0.05), diarrhoea (*p* < 0.05) (Figure 1B,C). To further evaluate whether Dio alleviated intestinal injury, we identified D-Lactate level in each group (Figure 1D). CDDP injection dramatically increased serum D-Lactate level, indicating severe intestinal damage and successful establishment of animal model (*p* < 0.01). D-lactate levels were decreased in Dio + CDDP rats compared with CDDP group (*p* < 0.01), which showed that Dio had protective activity. (*p* < 0.01). Moreover, the rats in the Dio-alone treated group showed no significant changes compared to control group, indicating that Dio had no obvious toxic side effects. 

### 2.2. Dio Attenuated CDDP-Induced Intestinal Histopathological Changes and Inflammatory Responses 

Histopathological changes in intestine are shown in Figure 2A. H and E staining’s clearly showed that in the control group, the intestinal villi appeared slender, the glands were intact, the nuclei were normal, and the epithelial cells were neatly arranged and columnar. In rats treated with CDDP, we observed damage to the mucosal cells of the intestinal epithelium, the disruption of the central lacteal duct, glandular deformations, and crypt cell apoptosis. All of these changes were considered severe intestinal injury (*p* < 0.01). In contrast, Dio supplementation for ten days reduced the degree of intestinal villous gland distortion, mucosal cells and crypt cells ablated in the intestinal epithelium (Figure 2B) (*p* < 0.01). In addition, Dio treatment can also alleviate the sharp decrease in intestinal villus height and shallowing of crypt depth (*p* < 0.01 and *p* < 0.05) (Figure 2C,D).

MPO activity is an important marker of neutrophile granulocyte infiltration. While NO synthesized by iNOS is related to various pathophysiological processes including inflammatory diseases in intestinal tract [25]. As shown in Figure 2E,F, MPO and iNOS activity were remarkably elevated in the CDDP-treated mucositis group compared to the control-treated group (*p* < 0.01). However, pretreatment with Dio significantly inhibited the elevated MPO and iNOS activity (*p* < 0.01 and *p* < 0.05 vs. CDDP). The results manifested a beneficial effect of Dio on CDDP-induced experimental mucositis via preventing inflammatory infiltration.

### 2.3. Dio Increased the Expression of Intestinal Mucin in Rats with Intestinal Mucositis

Goblet cells, important cells in epithelial lining in the intestine, contribute to innate immunity by secreting mucin glyco-proteins [26]. To confirm the protective effect of Dio on goblet cell secretion, in this experiment we investigated changes in the number of goblet cells of the intestinal mucosa by AB staining. As shown in Figure 3A, the ileum sections of the CDDP-treated group showed distorted crypts of Lieberkuhn, the presence of mucus at the apical surfaces of the sections and goblet cell disintegration compared with the control group. However, the Dio supplementation alleviated CDDP-induced distorted crypts of Lieberkuhn, mucus spillage, and maintained the number of goblet cells (Figure 3B) as compared to CDDP-treated group (*p* < 0.01). Notably, CDDP treatment significantly down-regulated the expression of mucin synthesis genes (MUC2 and MUC4) in the rats’ ilea (Figure 3C,D), while the expression levels of MUC2 and MUC4 mRNA in the ilea were increased by Dio pretreatment for CDDP-iv rats (*p* < 0.01 and *p* < 0.05). 

### 2.4. Dio Protected the Mucosal Barrier and Reduces Bacterial Translocation by Modulating TJs Proteins

At 72 h after CDDP treatment, the bacterial translocation of rat MLNs was examined. The CDDP-treated group presented elevated bacterial translocation in comparison with the control group (Figure 4A,B) (*p* < 0.01). Conversely, the over-translocations of bacteria were inhibited by Dio pretreatment in rats, indicating that Dio can reduce the permeability of the intestine to some extent, thereby protecting the integrity of the intestine. For further validating the integrity of the intestinal physical function, quantitative analysis from western blotting and real-time PCR were used to determine the expression levels of Claudin-1, ZO-1 and Occludin proteins. The results show that the protein and mRNA expression levels of Claudin-1, ZO-1 and Occludin decreased significantly in CDDP group (*p* < 0.01), suggesting that CDDP could cause damage to the tight junction structure of small intestine. However, Dio-pretreatment significantly rescued the reduction of these barrier proteins (*p* < 0.01) (Figure 4C–G).

### 2.5. Dio Inhibited TLR4-MyD88-NF-κB Signaling Pathway in Mucositis Rats

As depicted in Figure 5A–E, the protein expressions of TLR4, MyD88, p-IκBα and p-NF-κB were significantly augmented, whereas the expressions of IκBα and NF-κB were almost identical in the CDDP-treated group compared to the control group (all *p* < 0.01). However, pretreatment with Dio significantly decreased the protein expressions of TLR4 and MyD88, and reduced the ratios of p-IκBα/IκBα and p-NF-κB/NF-κB (*p* < 0.05 and *p* < 0.01). It has been noted that the dioscin pretreatment reduced the expression levels of TLR4, MyD88, IκBα and NF-κB mRNA in the intestine of CDDP-iv rats (*p* < 0.05) (Figure 5F). Activation of the TLR4-MyD88-NF-κB signaling pathway stimulates downstream inflammatory responses. As illustrated in Figure 5G–L, CDDP significantly increased the mRNA and protein expressions of pro-inflammatory cytokines, including TNF-α, IL-1β and IL-6 in rats, as compared to those of the control group (*p* < 0.01). CDDP treatment also significantly decreased the expression of anti-inflammatory cytokine IL-10 (*p* < 0.01). However, pretreatment with Dio significantly rescued these CDDP-induced changes.

These results manifested that pretreatment with Dio might down-regulate the protein expression of TLR4 and MyD88, and inhibit phosphorylation of IκBα and NF-κB, thereby suppressing the TLR4-MyD88-NF-κB signaling pathway and reducing the formation of inflammatory chemokines. This mechanism might contribute to regulating immune function to reduce inflammation and intestinal mucosal injury.

### 2.6. Dio Improved Gut Microbiota Dysbiosis in CDDP-Induced Intestinal Mucositis

Alterations to gut microbes comprise one of the basic features of the progression of intestinal inflammation in gastrointestinal mucositis [9]. To further evaluate the protective effect of Dio supplementation on cisplatin-induced intestinal mucositis, alterations to gut microbiota were investigated by 16S rRNA sequencing. The rarefaction curve of each sample tended to reach a saturation plateau (Appendix A), indicating that the sequencing depth covered rare phylotypes and most of the diversity. The result indicated that 1,213,614 usable reads and 3273 OTUs were obtained from 18 samples. As expected, the Venn diagrams showed three sets of overlapping OTU data, indicating that the 41.5% of OTU similarity between the control and CDDP treatment groups, and there was 49.4% of OTU similarity between the control-treated and Dio-pretreat rats (Figure 6A). The results of Anosim analysis are shown in Appendix A. Based on Bray-Curtis algorithm, the R value > 0, indicating that the gap between groups is greater than the gap within groups. NMDS and PCoA (weighted UniFrac analysis) plots of these data are shown in Figure 6B, a marked overall structural shift of gut microbiota in samples of CDDP-induced mucositis rats compared to the control group. Whereas the overall gut microbiota structure in the Dio + CDDP group was similar to that of the control group. In the system clustering tree (Appendix A), the phylogenetic relationship (UPGMA clustering tree) of the Dio + CDDP group was relatively close to the control-treated group on weighted UniFrac analysis. These results indicated that Dio pretreatment suppressed CDDP-induced changes in intestinal bacteria structure in rats with mucositis. Further, in contrast to that in the control group, CDDP reduced the α-diversity of intestinal bacteria (Figure 6C), which was shown by the decrease in the Chao, PD_whole tree, Shannon indexes (all *p* < 0.01). However, the decreases in bacterial community richness indexes were reversed by Dio supplementation (*p* < 0.05). 

In order to further evaluate the effects of CDDP treatment and Dio pretreatment on intestinal microbes in rats at different classification levels (phylum, family and genus), the amount of species were determined and analyzed statistically. The major intestinal bacteria at the phylum level included *Firmicutes*, *Proteobacteria and Bacteroidetes* (Figure 7A). Moreover, CDDP treatment decreased the relative abundance of *Firmicutes* and enriched the abundance of *Proteobacteria* compared to the control group; however, Dio treatment mitigated the CDDP-induced phylum-level changes. Figure 4B reveals the changes in abundances of genera among the three different groups. The CDDP group exhibited proportional decreases in the abundances of *Lachnospiraceae* and *Lactobacillus*, but these changes were attenuated by Dio administration (Figure 4B). In addition, the relative abundances of some pathogenic bacteria such as *Escherichia–Shigella* and *Parasutterella* significantly decreased in the Dio-pretreated group compared to those in the CDDP-treated group (Figure 7B). Species of interest (top 10 genera in each group by default) were selected to draw the classification tree, as shown in Figure 7C. After CDDP injection, *Enterobacteriales* was found to be enriched in the CDDP group at the order level. While Dio pretreatment can increase diversity and enhance the growth of *Lachnospiraceae* and *Ruminococcaceae* after CDDP treatment at the family level. It is worth noting that in the Dio+CDDP group, gut microbiota composition largely shifted to the genus *Blautia and Ruminiclostridium* (Figure 7C). In addition, taxonomic biomarkers were investigated, based on linear discriminative analysis effect size (LEfSe) and cladogram (circular hierarchical tree; Appendix A). Biomarkers from the three groups are shown in Figure 7D. Potentially enteropathogenic bacteria, such as *Escherichia coli* of the phylum *Proteobacteria*, were predominant biomarkers in the CDDP group. Genus *Blautia*, *Ruminococcaceae* NK4A214 group and family *Lachnospiraceae* of the phylum *Firmicutes* were the predominant biomarkers in the Dio+CDDP group (*p* < 0.01 and *p* < 0.05). Collectively, our results revealed that CDDP-induced gut microbiota dysbiosis was restored to a normal level by Dio treatment.

## 3. Discussion

Cisplatin has been studied as a broad-spectrum antitumor drug for more than 40 years. However, it is very cytotoxic and can cause a wide variety of adverse reactions in cancer patients. In recent years, remarkable progress has been made in effective protective regimens for ototoxicity and nephrotoxicity of CDDP [27,28]. However, gastrointestinal reactions after CDDP treatment are common and cause great pain for patients receiving chemotherapy. Research on intestinal toxicity and drugs’ improvement is still limited. In China, dioscin (a natural product) has been widely used in the clinical treatment of cardiovascular disease for many years. Researchers revealed that Dio pre-administration prevents many metabolic diseases and relieves acute damage to organs [29]. Furthermore, Dio has already been shown to prevent DSS-induced colitis in mice by enhancing intestinal barrier function and regulating macrophage polarization [30,31]. Therefore, we determined the effects of Dio on the intestine using a CDDP-induced mucositis rat model. In our study, we found that pretreatment with Dio significantly reduced the clinical symptoms of mucositis, such as weight loss and diarrhea. D-lactate is an intestinal bacterial metabolite produced by the intestine [32]. The content of D-lactate in plasma is a sensitive indicator for the early diagnosis of intestinal mucosal injury. The results showed that pretreatment with Dio could alleviate intestinal mucosal damage caused by CDDP. Increased intestinal MPO and iNOS activities are common features of chemotherapy-induced intestinal mucositis [33,34]. In the present study, histopathological assessment indicated that CDDP induced ablation of mucosal cells and crypt cells in the ileum, leading to increased intestinal permeability [35]. In contrast, pretreatment with Dio significantly alleviated tissue destruction caused by CDDP, improved histological scores, and decreased MPO and iNOS activities, alleviated mucosal inflammation. Moreover, no toxic reaction was observed in the Dio-treated rats, suggesting the relative safety for dioscin’s potential clinical application.

Chemotherapeutic agents can increase intestinal epithelial barrier permeability by inhibiting mucus secretion and decreasing protein expression of TJs, which is an important cause of intestinal mucositis [13,36]. In the present study, the phenomenon we observed in CDDP-induced rat mucositis is consistent with previous studies showing a decrease in the number of goblet cells, increased bacterial translocation, and decreased mucus mRNA and TJ protein levels. MUC2 and MUC4 are the characteristic secretory and membrane-bound mucins of gastrointestinal cells [37], which are the main components of the mucus barrier and are the first line of defense to prevent pathogen invasion. TJs (e.g., occludin, zonula occludens, and claudin-1)) play a role in establishing and maintaining intestinal tissue homeostasis by regulating barrier function, cell proliferation, and cell migration [38]. It was previously shown that Dio attenuated the disruption of barrier function in the colonic epithelium of mice with DSS-induced colitis [31]. Similarly, in the present study, Dio treatment could significantly restore the number of goblet cells and upregulated the mRNA and/or protein expression levels of MUC2, MUC4, ZO-1, Occludin and Claudin-1 in the ileum. These results demonstrated that Dio can avoid bacterial invasion by protecting the intestinal barrier function.

Activation of Toll-like receptors (TLRs) has a key role in regulating intestinal innate immunity. TLR4 mediates immune and inflammatory responses to microbial infection [39]. It has been demonstrated that TLR4 is activated and recruits MyD88 during the disruption of intestinal homeostasis, which triggers the NF-κB signaling cascade, increases proinflammatory cytokines, and aggravates the inflammatory response to further damage the mucosa [40]. Our data suggested that pretreatment with Dio dramatically inactivated TLR4, MyD88, phosphorylated IκBα and NF-κB p65. In contrast, Dio efficaciously suppressed the expression of NF-κB downstream pro-inflammatory cytokines, including TNF-α, IL-6 and IL-1β in the ileums of CDDP-induced mucositis rats, and substantially promoted the production of the immunoregulatory mediators IL-10. These results proposed that inhibition of the TLR4-linked NF-κB signaling pathway might play a crucial role in the protection of Dio against CDDP-induced mucositis. 

Several recent studies have shown that gut bacteria can modulate TJ expression and assembly, thereby regulating transepithelial permeability [11,41]. In addition, it has been pointed out that transgenic expression of unmodified TLR4 is associated with changes in microbial density and composition and alterations of the intestinal barrier [42]. An unstable balance between the gut microbiota and the host may contribute to a variety of diseases characterized by gut barrier disruption [43,44]. Therefore, maintaining healthy and stable gut microbiota may serve as an effective way to prevent chemotherapy-induced mucositis. 

Evidence is growing that the gut microbiota modulate the host response to chemotherapeutic drugs. The gut microbiota can now, therefore, be targeted to improve efficacy and reduce the toxicity of current chemotherapy agents [45]. In addition, it has already been shown that CDDP induces significant changes in the intestinal commensal bacterial repertoire, thereby exacerbating mucosal damage. This stems from the fact that CDDP has antibiotic activity against some bacteria [46]. Therefore, restoration of the microbiota drives healing of cisplatin-induced intestinal damage [35,46]. In addition, the metabolism of many drugs is known to be affected by the gut microbiome [47], and Dio is no exception [48]. More evidence now points to the effect that edible biologic medicine can have on the function and composition of the gut microbiome [49]. In this study, Dio modulated gut microbiota structure, restored its diversity, and maintained gut community structure uniformity in CDDP-induced mucositis rat (Figure 6). The results also clearly showed that gut microbiota were severely imbalanced in CDDP-induced mucositis rats, as shown by a dramatic increase in the abundance of the phyla *Proteobacteria* (mucosa-associated inflammation-promoting bacteria) and a decrease in the abundance of *Firmicutes*, which is consistent with the gut bacterial structure of cancer patients who develop gastrointestinal mucositis after clinical chemotherapy [35]. However, Dio pretreatment significantly reverted the *Firmicutes* to *Proteobacteria* ratio (Figure 7A). It has been demonstrated that the number of *E. coli* in the ileum is related to the severity of ileitis in patients [50]. In our study, CDDP treatment also led to an increase in the abundance of *E. coli*, however, this increase was attenuated with Dio treatment. Clinical studies have confirmed that *Lachnospiraceae* and *Ruminococcaceae* microorganisms play a role in alleviating intestinal inflammation and repairing the intestinal barrier, acting as protective intestinal commensal strains [51,52], and the most important explanation for this is that they are responsible for degrading various polysaccharides and fibers and producing short-chain fatty acids (SCFAs) [53,54]. SCFAs are important substrates for maintaining intestinal epithelial cells and regulating the immune system and inflammatory response [55,56]. In this study, Dio treatment promoted the growth of members of the *Lachnospiraceae* and *Ruminococcaceae*. Therefore, it may be deduced that part of Dio was degraded and fermented by *Lachnospiraceae* and *Ruminococcaceae*, which is beneficial to its growth and advantages, thereby reducing mucosal damage.

In conclusion, the rat model of chemotherapy mucositis was successfully induced using CDDP in this paper. The current results suggest that Dio effectively alleviates CDDP-induced intestinal injury in rats by attenuated gut microbiota dysbiosis, repairing the intestinal barrier and suppressing the TLR4-MyD88-NF-κB signaling pathway. Taken together, these results provide a reference and can be used as a benchmark for the development of cancer treatment strategies.

## 4. Materials and Methods

### 4.1. Induction of Mucositis Model and Experimental Design

Female Wistar rats (6 weeks old), weighing 176–200 g, were purchased from CHANGSHENG Experimental Animals Co. Ltd. (Changchun, Jilin province, China), and were housed in cages at a controlled temperature (25 ± 2 °C) and humidity (60 ± 10%) with 12 h light/dark cycle. Rats were divided randomly groups (3 per cage) and were fed ad libitum with standard food and provided fresh tap water. Following 1 week of acclimatization, rats were randomly divided into four groups (6 rats per group): control group (0.5% CMC-Na), Dio only group (60 mg/kg), CDDP group (6 mg/kg), CDDP combined with Dio group. Dioscin (C_45_H_72_O_16_, MW: 869.05, CAS: 19057-60-4, HPLC purity ≥ 98%) was obtained from DESITE Biological Technology Co., Ltd. (Chengdu, China). Cisplatin was purchased from Sigma–Aldrich Co. (St. Louis, MO, USA). Carboxymethyl cellulose sodium (CMC-Na) was purchased from JinPin Chemical technology Co., Ltd. (Shanghai, China). 

As shown in Figure 1A, Dio was dissolved in 0.5% CMC-Na and intragastrically administered once a day for 10 consecutive days. While a single injection of CDDP was administered into a tail vein after the 7th day of administration of cisplatin. In current study, the doses of CDDP and Dio were selected based on previous published literature [20,22,57]. Body weight was monitored daily after acclimatization period. Seventy-two hours after administration of cisplatin, rats from all groups were euthanized (via exsanguinations) under sodium pentobarbital. Blood was collected and centrifuged twice at 3000× *g* for 10 min, and serum was separated and stored in an ultra-low temperature freezer at −80 ℃ until testing. At the same time, mesenteric lymph nodes (MLNs), cecal contents, and the tissues (7–15 cm segment before the cecum, the most representative parts of the degree of intestinal damage) were aseptically harvested and frozen in liquid nitrogen before storage at −80 ℃ for the next analysis. All animal experiments were approved by the Institutional Animal Care and Use Committee of the Northeast Agricultural University under the approved protocol number SRM-11.

### 4.2. Diarrhoea Assessment

Diarrhoea was measured using the mean scores, and its severity was quantified according to the procedure described by Kurita et al. [58]. As follows: 0—normal (normal stool or the absence of stool); 1—slight (slightly wet and soft stool); 2—moderate (wet and unformed stool with moderate perianal staining of the coat); and 3—severe (watery stool with severe perianal staining of the coat).

### 4.3. Assessment of Biochemical Parameters

The concentration of D-Lactate in peripheral blood was measured using ELISA kits according to the manufacturers’ instructions (JINGMEI Bioengineering Co., Ltd. Yancheng, Jiangsu, China). The levels of iNOS and MPO in intestinal tissues were detected according to the manufacturer’s instructions (JianCheng Biotechnoloy Co., Ltd., Nanjing, China). INOS can catalyze the reaction of L-Arg and molecular oxygen to generate NO, and NO generates colored compounds with nucleophiles. INOS activity in the exiting intestinal tissue can be detected by calculating the magnitude of absorbance at a wavelength of 530 nm. MPO is an enzyme specific to neutrophils and has the ability to reduce hydrogen peroxide. It generates a yellow compound after hydrogen supply by the hydrogen donor o-dianisidine, and the generation of MPO can be determined by calculating the absorbance at 460 nm wavelength.

### 4.4. Histopathological Analysis of the Ileum

Ileum tissues were fixed with Carnoy’s fluid (Lanji Biotechnoloy Co., Ltd., Shanghai, China) for 24 h, embedded in paraffin. Sections of 5 μm were stained with conventional hematoxylin & Eosin (H&E) and Alcian blue (AB)-nuclear fast red (LEAGENE Biotechnoloy Co., Ltd., Beijing, China). After these procedures, sections were scanned with a fully automated digital slide scanning system HN-610. The caseviewer program was used for image capturing and the morphometric parameters were captured under 160–450 × fields of view. The experimenter was unaware of the experimental treatments.

Histological damage was evaluated in sections stained with H&E using criteria adapted from Galeazzi et al. [59]. A numerical score of 0–9 was assigned to each section considering general loss of mucosal architecture (graded 0–3, absent to severe), extent of inflammatory cell infiltrate (graded 0–3, absent to transmural), crypt cells ablation (0–1, absent or present), goblet cell depletion (0–1, absent or present), and muscular layer thickness (0–1, normal to reduced). The intestinal villus height and crypt depth were also measured. After AB staining, the mean integral optical density of goblet cells was determined using Image-Pro Plus (IPP) image software. 

### 4.5. Bacterial Translocation Assay 

We took 100 mg of MLN from each rat and added 400 μL of PBS buffer. After mixing, it was placed in a tissue homogenizer, ground and centrifuged (4 °C, 650 rpm, 3 min). Then, 100 μL of supernatant was aspirated and plated onto Brian Heart Infusion agar (Hope Biotechnoloy Co., Ltd., Qingdao, China). After 24 h incubation at 37 °C, the number of colonies on the plate was recorded.

### 4.6. Western Blotting Assay

Frozen ileum tissues were cut into small pieces and lysed with RIPA lysis buffer (Beyotime Biotechnology, Shanghai, China). Phenylmethanesulfonyl fluoride (PMSF) (Beyotime Biotechnology, Shanghai, China) was added, and the tissue was homogenized through a Tissue Grinding instrument (Shanghai Jingxin Industrial Development Co., Ltd., Shanghai, China), and then centrifuged at 3000 rpm for 10 min at 4 °C to collect the supernatant. Protein concentrations were quantified by a BCA Protein Assay kit (Beyotime Biotechnology, Shanghai, China). Equal amounts of protein sample were separated by standard Tris-glycine SDS-PAGE gel electrophoresis, and then transferred to polyvinylidene difluoride (PVDF) membranes. After blocking with 5% skimmed milk for 2 h at room temperature, the PVDF membranes were incubated with primary antibodies at 4 °C overnight. Primary antibodies and dilutions were as follows: ZO-1 (WL03419, Wanlei, Shenyang, China) diluted 1:1000; Occludin (WL01996, Wanlei, Shenyang, China) diluted 1:1000; Claudin-1 (WL03073, Wanlei, Shenyang, China) diluted 1:1000; TLR4 (WL00196, Wanlei, Shenyang, China) diluted 1:1000; MyD88 (WL02494, Wanlei, Shenyang, China) diluted 1:500; NFκΒ (WL01980, Wanlei, Shenyang, China) diluted 1:500; p-NFκΒ (WL02169, Wanlei, Shenyang, China) diluted 1:1000; IκΒα (WL01936, Wanlei, Shenyang, China) diluted 1:500; p-IκΒα (WL02495, Wanlei, Shenyang, China) diluted 1:500; IL-1β (WL02385, Wanlei, Shenyang, China) diluted 1:500; IL6 (WL02841, Wanlei, Shenyang, China) diluted 1:1000; IL10 (WL03088, Wanlei, Shenyang, China) diluted 1:1000; TNF-α (WL01581, Wanlei, Shenyang, China) diluted 1:1000; β-tubulin (WL01931, Wanlei, Shenyang, China) diluted 1:1000. After washing five times with Tris-buffered saline containing Tween (TBST), the membranes were incubated with 1:10,000 horseradish peroxidase-conjugated rabbit anti-mouse IgG secondary antibody (WLA024, Wanlei, Shenyang, China) at room temperature for 2 h and then washed with TBST, followed by development using ECL reagent (Beyotime Institute of Biotechnology, Shanghai, China), captured by the Amersham Imager 600 software (GE, Boston, MA, USA), and analyzed using Image J software.

### 4.7. Quantitative Real-Time Polymerase Chain Reaction (PCR) Assay 

Total RNA was isolated from the ileum tissue with Trizol reagent (Invitrogen, Carlsbad, CA, USA) according to the manufacturer’s instructions, and reverse transcribed into cDNA using the PrimeScript RT reagent kit (DRR037A; Takara, Dalian, China). Then, quantitative real-time PCR detection of RNA copies were performed on a Light Cycler^®^ 480 II Detection System (Roche, Basel, Switzerland) using IQ SYBR Supermix reagent (Bio-Red, San Diego, CA, USA). The relative expression levels were normalized to GAPDH and analyzed by the 2^−ΔΔCt^ method. The list of primers for the detection of mRNA expression are listed in Appendix A

### 4.8. Gut Microbiota Analysis

Six different cecal contents of each group were randomly collected and stored at −80 °C. The metagenomic DNA was extracted using CTAB/SDS method. Hypervariant V4 region of bacterial 16S Journal Pre-proof 11 rRNA gene was amplified using the primers 515F (: 5′-GTGCCAGCMGCCGCGGTAA-3′) and 806R (5′-GGATACHVGGGTWTCTAA-T-3′). Library preparation and sequencing was carried out after PCR products mixing and purification. Sequencing libraries were generated using TruSeq^®^ DNA PCR-Free Sample Preparation Kit. The library quality was assessed on the Qubit@ 2.0 Fluorometer (version 2019.4). At last, the library sequencing was performed on Illumina HiSeq 2500 platform.

Quality filtering on the raw reads were performed to obtain the high-quality clean reads according to the Qiime (V1.9.1, http://qiime.org/scripts/split_libraries_fastq.htm, accessed on 28 January 2021) quality-controlled process. The reads were compared with the reference database (Silva database, https://www.arb-silva.de/ accessed on 28 January 2021) using UCHIME algorithm (UCHIME Algorithm, http://www.drive5.com/usearch-/manual/uchime_algo.html accessed on 28 January 2021) to detect chimera sequences, and then the chimera sequences were removed. Species composition was revealed by reads shear filtration, operational taxonomic units (OTUs) clustering (Uparse v7. 0.1001, http://drive5.com/uparse/ accessed on 28 January 2021, sequences with ≥97% similarity were assigned to the same OTUs), species annotation (the Silva Database, https://www.arb-silva.de/ accessed on 28 January 2021 was used based on Mothur algorithm to annotate taxonomic information), and abundance analysis. OTUs abundance information were normalized using a standard of sequence number corresponding to the sample with the least sequences. Subsequent analysis of alpha diversity and beta diversity were all performed basing on this output normalized data. Alpha diversity was also estimated using the phylogenetic diversity metric. Beta diversity analysis was used to compare gut microbiota compositions among the groups and was performed using the unweighted pair group method with arithmetic mean (UPGMA) clustering method based on weighted and unweighted UniFrac distances. Two-dimensional non-metric multidimensional scaling (NMDS) and principal coordinate analysis (PCoA) plots with a weighted uniFrac matrix were used to assess the variation (β-diversity distance) between experimental groups. Different taxa microbes were identified based ontaxon-based analysis and LEfSe analysis.

### 4.9. Statistical Analysis

All the data were analyzed using the SPSS 22.0 statistical package (SPSS Inc., Chicago, IL). Comparisons among different experimental groups were performed using a one-way analysis of variance (ANOVA) and the Tukey-Kramer test. Data were reported as mean ± standard deviation (M ± SD). In addition, the Kruskal Wallis test in SPSS software was used for statistical testing of diarrhea score and histological damage score data. The data was plotted using GraphPad Prism 7 (GraphPad Software, La Jolla, CA, USA). A value of *p* < 0.05 or *p* < 0.01 was considered as statistically significant.

## Figures and Tables

**Figure 1 ijms-23-04431-f001:**
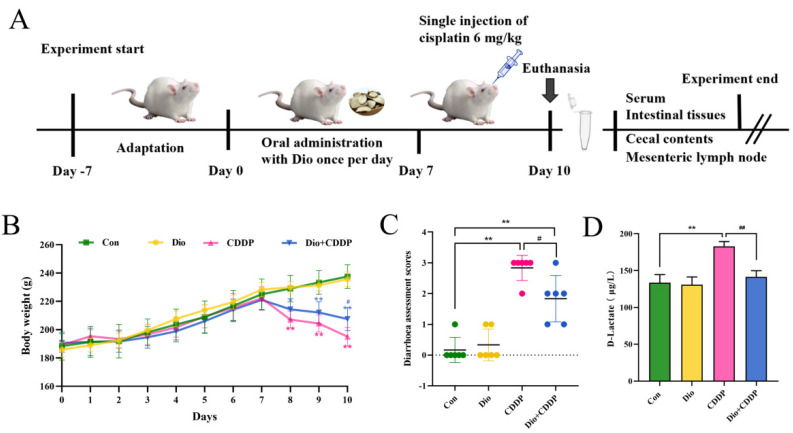
Dio ameliorated the symptoms of CDDP-induced mucositis in rats. (**A**) Schematic diagram of the experimental design. (**B**) Daily bodyweight changes from day 1 to 10. (**C**) Diarrhoea assessment score. (**D**) D-Lactate level. All values were expressed as Mean ± S.D. (n = 6 samples per group). * *p* < 0.05, ** *p* < 0.01 vs. control group, # *p* < 0.05, ## *p* < 0.01 vs. CDDP group.

**Figure 2 ijms-23-04431-f002:**
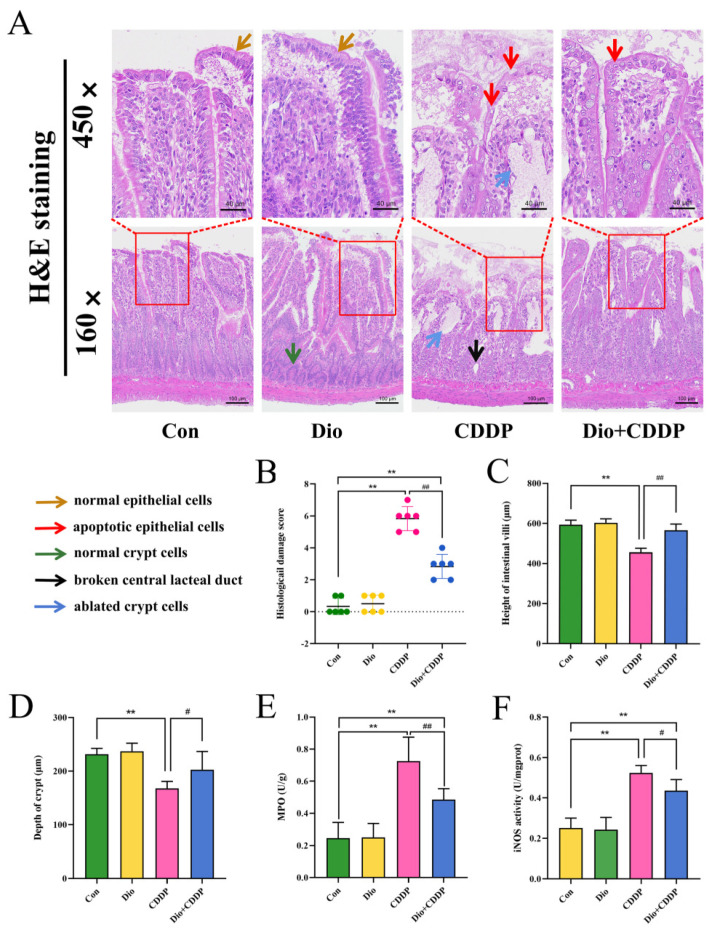
Ameliorative effect of Dio on ileum tissue injury and inflammatory cell infiltration in rats with CDDP-induced mucositis. (**A**) Intestinal tissues were stained with H and E dye kits (160×, 450×). Note: Normal epithelial cells (Yellow arrows), apoptotic epithelial cells (Red arrows), normal crypt cells (blue arrows), and Black and blue arrows indicate ablated crypt cells and broken central lacteal duct in cisplatin rats, respectively. (**B**) Histological damage score. (**C**) villus length, (**D**) crypt depth. In addition, determination of MPO (**E**) and iNOS (**F**) activity to determine inflammatory infiltration. All values were expressed as Mean ± S.D. (n = 6 samples per group). ** *p* < 0.01 vs. control group, # *p* < 0.05, ## *p* < 0.01 vs. CDDP group.

**Figure 3 ijms-23-04431-f003:**
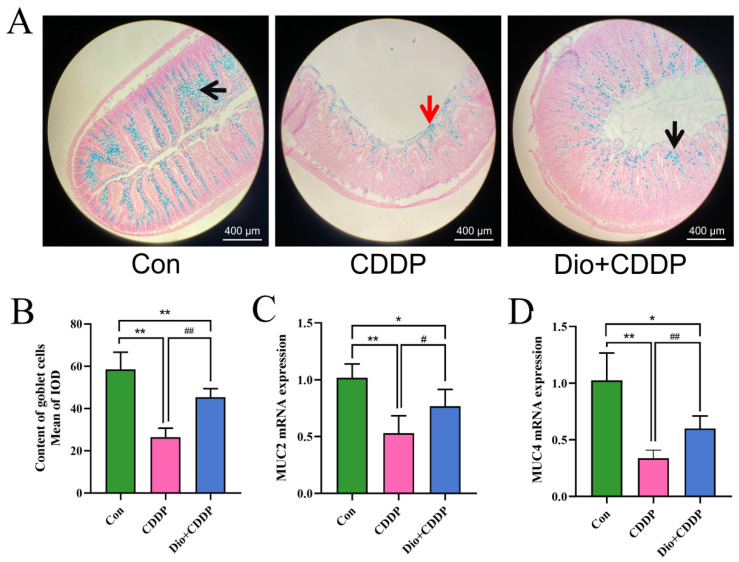
Dio administration promoted the expression of mucins and enhanced the quantity of goblet cells in ileum. AB- Nuclear fast red staining of small intestine sections in the 3 groups of rats (**A**), black arrows indicate the mucin glycoproteins in goblet cells, red arrows indicate mucus spillage. Quantitative analysis of goblet cell in intestinal tissue (**B**). Effect of Dio on the mRNA levels of MUC2(**C**) and MUC4 (**D**). All values were expressed as Mean ± S.D. (n = 6 samples per group). * *p* < 0.05, ** *p* < 0.01 vs. control group, # *p* < 0.05, ## *p* < 0.01 vs. CDDP group.

**Figure 4 ijms-23-04431-f004:**
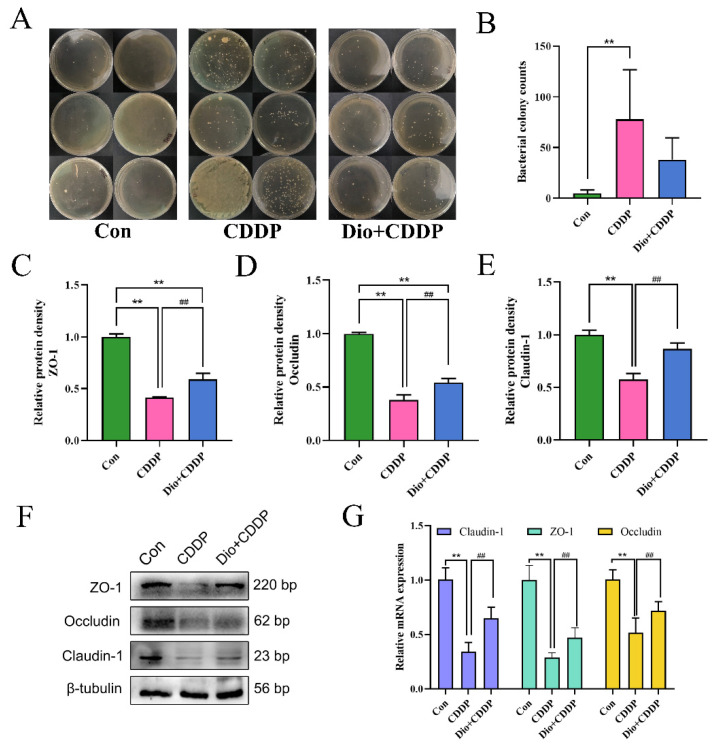
Dio protects the intestinal physical barrier by inhibiting CDDP-induced loss of E-cadherin and TJs proteins expression. (**A**) Bacterial culture results of MLNS. (**B**)Bacterial translocation was analyzed by counting the number of colonies on the plate. (**C**–**E**) Changes in the relative protein expression levels of ZO-1, Occludin and Claudin-1 were measured respectively. (**F**) Representative Western blotting images of E-cadherin and TJs proteins, and the relative protein expressions were normalized to β-tubulin. (**G**) Relative mRNA expression of Claudin-1, ZO-1 and Occludin (n = 6 samples per group). ** *p* < 0.01 vs. control group, ## *p* < 0.01 vs. CDDP group.

**Figure 5 ijms-23-04431-f005:**
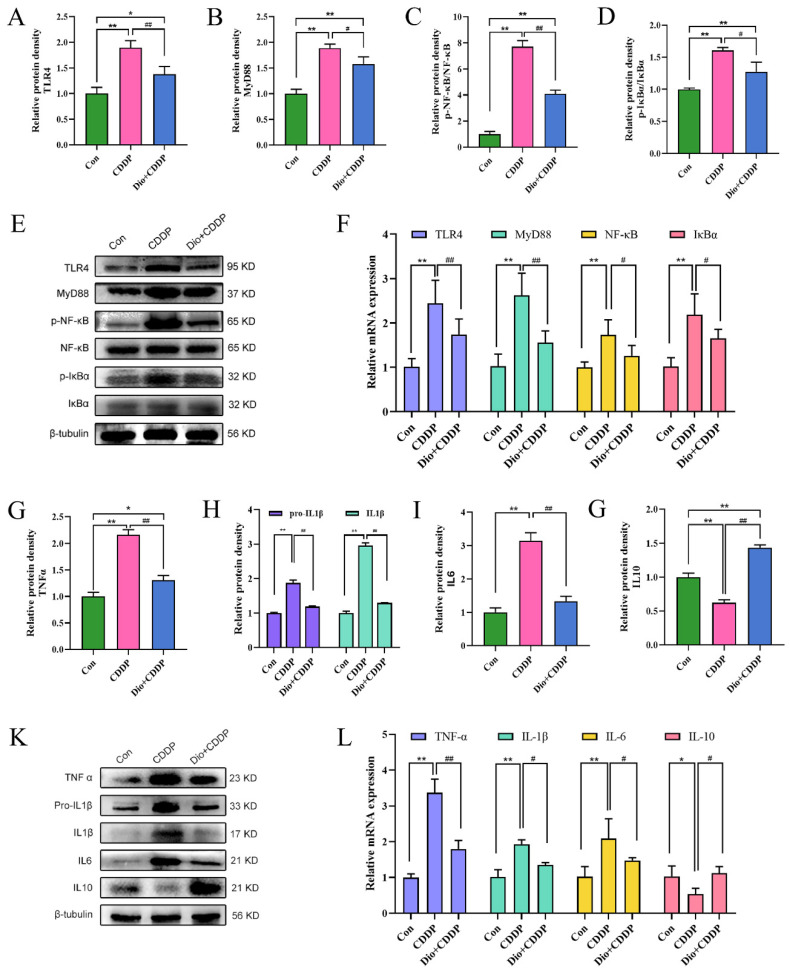
Effect of Dio on the activation of TLR4-MyD88-NF-κB signaling pathway and immune-inflammation status in rats with CDDP-induced mucositis. Changes in the relative protein expression levels of TLR4 (**A**), MyD88 (**B**), p-NF-κB/NF-κB ratio (**C**) and p-IκBα/IκBα ratio (**D**). (**E**) Representative Western blotting images of TLR4, MyD88, p-NF-κB, NF-κB, p-IκBα and IκBα. (**F**) Relative mRNA expression of TLR4, MyD88, NF-κB and IκBα. Changes in the relative protein expression levels of TNFα (**G**), Pro-IL1β & IL1β (**H**), IL6 (I) and IL10 (**J**). (**K**) Representative Western blotting images of TNFα, Pro-IL1β, IL1β, IL6 and IL10. (**L**) Relative mRNA expression of TNF-α, IL-1β, IL-6 and IL-10. * *p* < 0.05, ** *p* < 0.01 vs. control group, # *p* < 0.05, ## *p* < 0.01 vs. CDDP group.

**Figure 6 ijms-23-04431-f006:**
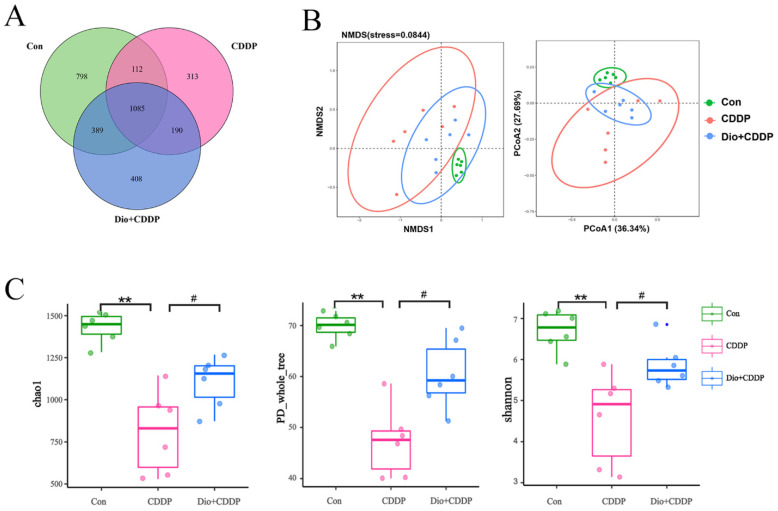
Effects of Dio administration on gut microbiota in CDDP-induced mucositis rats. (**A**) Venn diagram showing the overlap of OTUs identified in the gut microbiota among the three groups. (**B**) NMDS analysis and PCoA analysis of the three groups based on weighted UniFrac distances. Each plot represents one sample. (**C**) α-Diversity indicated by the chao index, PD_whole_tree index, and Ace index. All values were expressed as Mean ± S.D. ** *p* < 0.01 vs. control group, # *p* < 0.05 vs. CDDP group.

**Figure 7 ijms-23-04431-f007:**
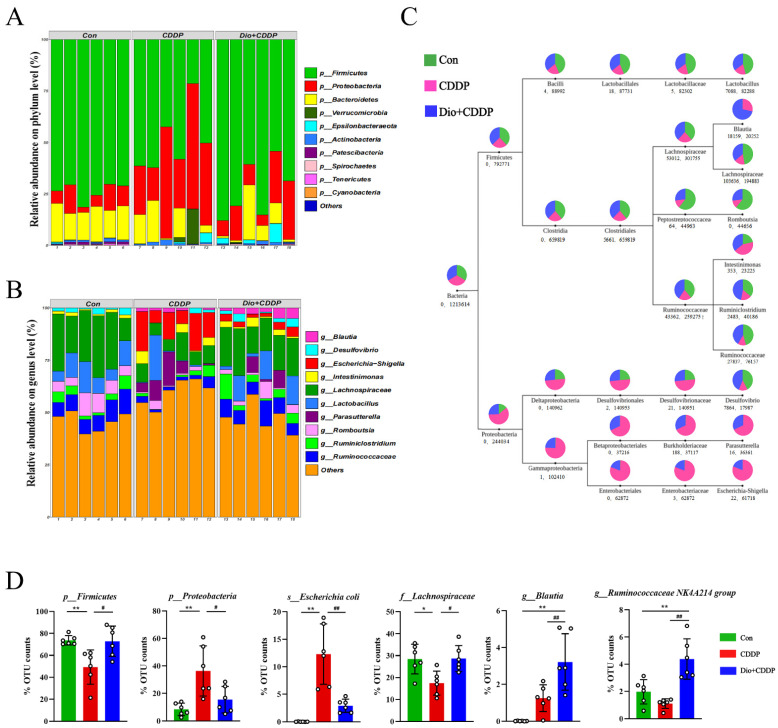
Taxonomic shifts in the microbes. (**A**) The proportion of dominant communities at the phylum level in each group of samples. (**B**) The proportion of dominant communities at the genus level in each group of samples. (**C**) Classification tree. The first circle represents the kingdom level (bacteria). The second circle (in same column) represents the phylum level. The subsequent order is class, order, family and genus. The number below the circle, the first indicates the number of sequences aligned to that classification only, and the second number indicates how many sequences are aligned to that classification in common. (**D**) Biomarkers are represented by the bar chart. The relative abundance with statistical differences biomarkers were compared among groups, including *p-Firmicutes*, *p-Proteobacteria*, *s-Escherichia coli*, *f-Lachnospiraceae*, *g-Blautia*, *g-Ruminococcaceae* NK4A214. * *p* < 0.05, ** *p* < 0.01 vs. control group, # *p* < 0.05, ## *p* < 0.01 vs. CDDP group.

## Data Availability

All data used during the study appear in the submitted article.

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
