# Peer review of "Dioscin Alleviates Cisplatin-Induced Mucositis in Rats by Modulating Gut Microbiota, Enhancing Intestinal Barrier Function and Attenuating TLR4/NF-κB Signaling Cascade"

_ijms, 2022, doi:10.3390/ijms23084431_

Round 1
Reviewer 1 Report
Thankyou for the opportunity to review this paper. It is on the whole well written, and it was interesting to see the clear effects of dioscin on cisplatin-induced mucositis.
I have some minor comments and suggestions:
- There are some relatively minor issues with grammar and use of tenses that could be improved.
- I found it at times difficult to understand what method or assay had been used to give a particular result. For example, it is not clearly listed in the methods or results section exactly what type of assay was used for the iNOS and MPO results. Including this information would help to better understand and clarify the results.
- Diarrhoea scoring and H&E scoring are graphically represented as mean ± S.D, with accompanying statistics. However both of these scores are ordinal, not continuous data, and therefore would ideally be displayed with a median, with statistical testing done using a Kruskal Wallis test. It is likely that this test will not alter the statistical significance very much, however it would be best practice to display and statistically test the data this way.
- There are some minor inconsistencies or missing details between figures that could be improved. For example, in figure 3B, is the number of goblet cells lister per crypt, or per a particular sized section of tissue?
- In the discussion, it would be useful to include some information around whether dioscin has been used successfully in clinical studies, and how dioscin may be used for cisplatin-induced mucositis in people. In particular, whether the dosing regimen used in this paper would be useful in a human population.
- Could you speculate on whether the microbial changes seen in this model correlate with the type of bacteria that are altered in the human gut following cisplatin treatment?
Author Response
Reply to reviewer 1 comment
Dear Reviewer 1
Thank you for your time and effort reviewing the previous version of the manuscript. Your suggestions allow us to improve our work. Following the instructions provided in your letter, we have uploaded the revised manuscript file. Any revisions made to the manuscript have been flagged using the Track Changes feature. Therefore, we have highlighted significant content changes in red text, corresponding to the reviewer's comments. Attached to this letter is our point-by-point response to the comments made by the reviewers. Comments are copied and our responses are then given directly in a different color (red). We would also like to thank you for your review of our resubmitted manuscript.
Point 1: There are some relatively minor issues with grammar and tense usage that could be improved.
Response 1: We apologize for the minor issues with grammar and tense usage in the manuscript. This manuscript has been checked by a native English-speaking colleague.
Point 2: I find it sometimes difficult to understand what method or assay to use to give a particular result. For example, the type of assay used for iNOS and MPO results was not explicitly listed in the Methods or Results section. Including this information will help to better understand and clarify the results.
Response 2: Thank you for highlighting this flaw. We have added the correct methods and assays in Materials and methods (lines 119 and 141).
Point 3: Diarrhea scores and H&E scores are graphically presented as mean ± SD with statistics. However, both scores are ordered, not continuous data, so ideally would be shown as median, and statistically tested using the Kruskal Wallis test. The test probably won't have much impact on statistical significance, but it would be best practice to display and count test data this way.
Response 3: Thank you for introducing us to the Kruskal Wallis test method. We performed statistical tests on the diarrhea score and pathology score data (line 220) using the Kruskal-Wallis test in SPSS software. And agree with the comments of Diarrhea score and H&E score, we changed the histogram of these two data to the form of scatter plot, which can reflect the score value more intuitively. (Figure 1C, Figure 2B)
Point 4: There are some minor inconsistencies or missing details between the numbers that could be improved. For example, in Figure 3B, what is the number of goblet cells per crypt or per tissue section of a particular size?
Response 4: We really appreciate the errors you pointed out. The number of goblet cells in the manuscript is not scientific enough. After consideration, we decided to use Image-Pro Plus (IPP) imaging software to count the mean integrated optical density of goblet cells (n=3) (Fig. 3B). Can you see if it works?
Point 5: In the Discussion, it would be useful to include some information on whether dioscin has been used successfully in clinical studies and how dioscin has been used to treat cisplatin-induced mucositis in humans. In particular, whether the dosing regimen used in this article is useful in the population.
Response 5: Thank you for your suggestion. We have added a discussion of the current status of clinical use of diosgenin (line 412). Unfortunately, Dio has been reported in the prevention of metabolic diseases but not in the clinical treatment of gastrointestinal diseases.
Point 6: Can you speculate whether the microbial changes seen in this model correlate with the types of bacteria that occur in the human gut after cisplatin treatment?
Response 6: We are very grateful to the reviewer for pointing out this issue. We revisited the association between the results of gut microbiota changes in the manuscript and chemotherapy-induced mucositis. (lines 482 and 488)
Briefly, some changes in gut bacteria in CDDP-treated rats (increased numbers of Enterobacteriaceae and decreased numbers of Firmicutes and Proteobacteria) were consistent with changes in gut bacteria in patients with chemotherapy-induced mucositis. Premedication of Dio mitigated these changes. In addition, Dio administration can promote the proliferation of the recognized protective bacteria Lachnospira and Ruminococcus.
If there are still some mistakes, I hope you will point them out, and we will definitely correct them.
Sincerely,
Liu Yun, E-mail: [email protected]
Jin Shengzi, E-mail: [email protected]
Reviewer 2 Report
After reviewing the manuscript by Jin et al., titled, “Dioscin alleviates cisplatin-induced mucositis in rats by modulating gut microbiota, enhancing intestinal barrier function and attenuating TLR4/NF-κB signaling cascade”, there are no major comments that require attention in the present version of the manuscript. The manuscript can be recommended for acceptance to publication. However, a careful editing of English language is recommended before final acceptance.
Author Response
Dear reviewer
We appreciate your time and effort in reviewing previous versions of your manuscript. We are glad that our research has been recognized by you. In accordance with the instructions provided in your letter, we have uploaded the files of the revised manuscript, the English manuscript has been carefully revised by our native English-speaking colleagues. We would also like to thank you for your review of our resubmitted manuscript.
Sincerely,
Yun Liu, E-mail: [email protected]
Shengzi Jin, E-mail: [email protected]